# A Bayes-Optimal View on Adversarial Examples

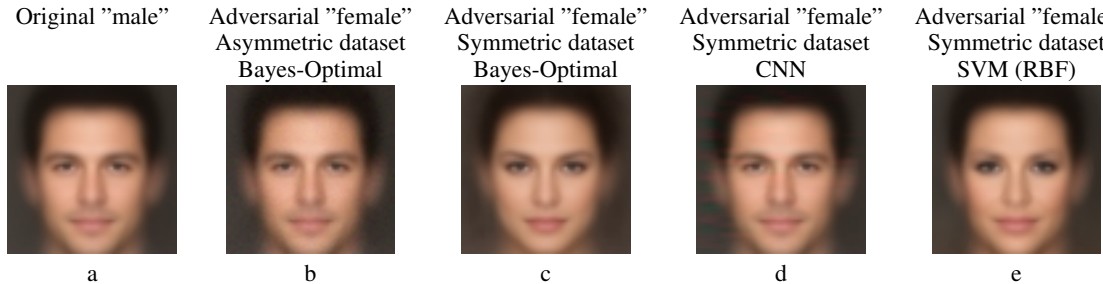

Figure 1: We analyze conditions on the data distribution under which the Bayes-Optimal classifier may be vulnerable to adversarial attacks and present realistic image datasets for which the Bayes-Optimal classifier can be calculated efficiently. When the data distribution satisfies certain asymmetries, the Bayes-Optimal classifier is vulnerable (b), but when the distribution is symmetric, the optimal classifier is robust and adversarial attacks are perceptually meaningful (c). Our experiments with these datasets show that CNN training consistently fails to find a robust classifier even when the optimal classifier is robust (d), while large-margin methods often succeed (e).

## Abstract

Adversarial attacks on CNN classifiers can make an imperceptible change to an input image and alter the classification result. The source of these failures is still poorly understood, and many explanations invoke the "unreasonably linear extrapolation" used by CNNs (Goodfellow et al., 2018) along with the geometry of high dimensions.

In this paper we show that similar attacks can be used against the *Bayes-Optimal* classifier for certain class distributions, while for others the optimal classifier is robust to such attacks. We present analytical results showing conditions on the data distribution under which all points can be made arbitrarily close to the *optimal* decision boundary and show that this can happen even when the classes are easy to separate, when the ideal classifier has a smooth decision surface and when the data lies in low dimensions. We introduce new datasets of realistic images of faces and digits where the Bayes-Optimal classifier can be calculated efficiently and show that for some of these datasets the optimal classifier is robust and for others it is vulnerable to adversarial examples. In systematic experiments with many such datasets, we find that standard CNN training consistently finds a vulnerable classifier even when the optimal classifier is robust while large-margin methods often find a robust classifier with the exact same training data. Our results suggest that adversarial vulnerability is not an unavoidable consequence of machine learning in high dimensions, and may be the result of the specific distributions of commonly used datasets or of suboptimal training methods used in current practice.

## 1 Introduction

Perhaps the most intriguing property of modern machine learning methods is their susceptibility to adversarial examples (Szegedy et al., 2014; Goodfellow et al., 2018): for many powerful classifiers it

is possible to change the input by an imperceptible amount and change the decision of the classifier. While adversarial examples were most famously reported for Neural Network classifiers (Szegedy et al., 2014), subsequent research has shown that other classifiers can also fall prey to similar attacks (Goodfellow et al., 2018). Attempts to make classifiers robust to these attacks have generated a tremendous amount of interest (e.g. (Schott et al., 2019) and references within), although a recent survey argued that "There has been much work showing that basically all defenses suggested so far in the literature do not substantially increase robustness over undefended neural networks." (Schott et al., 2019)

As a first step towards solving the problem, many authors have attempted to understand the source of the failure (Goodfellow et al., 2018; Szegedy et al., 2014; Tanay & Griffin, 2016; Fawzi et al., 2018). Intuitively the failure suggests that the decision surface learned by neural networks is "discontinuous to a significant extent", analogous to an attempt to discriminate the rational numbers from the rest of the real numbers (Szegedy et al., 2014). The most prominent theory suggests that the problem is that neural network classifiers are "unreasonably linear" combined with the fact that they operate in high dimensions (Goodfellow et al., 2018; 2015). In high dimensions the output of a random linear classifier can be changed by making a small change in the $\ell_\infty$ norm of an example. (Fawzi et al., 2018) show a connection between the error rate of linear and quadratic classifiers and the adversarial vulnerability and show that linear classifiers in high dimensions must be vulnerable for data that is not linearly separable. (Ford et al., 2019) show that adversarial vulnerability is closely related to the lack of generalization to random perturbations and that in high dimensions even moderate failures to generalize to high amounts of noise imply the existence of adversarial examples. Shamir et al. (2019) have also focused on the geometry of high dimensions arguing that adversarial attacks may be a "natural consequence of the geometry of $R^n$ with the $L_0$ (Hamming) metric". Jacobsen et al. (2018) connect the problem of adversarial examples with the excessive invariance of modern CNNs to large image transformations. Zhang et al. (2019) discuss the tradeoff between robustness and accuracy and show a very discontinuous toy distribution where the Bayes-Optimal classifier is not robust. Most recently, (Ilyas et al., 2019) have argued that adversarial examples are a feature, not a bug, and showed that one can in fact obtain information about the true decision boundary from adversarial examples. Their analysis suggests that vulnerability results from the presence of predictive features that are not robust. They presented a synthetic dataset which was constructed to not contain such features, and showed that CNN training on that dataset was robust. . In (Nakkiran, 2019), a similar dataset was presented without nonrobust features, but CNN training did lead to non robust classifiers when the dataset was noisy, leading the authors to conclude that "Adversarial Examples are Just Bugs, Too". This result is consistent with Tanay & Griffin (2016) who show that adversarial vulnerability is related to overfitting in learning algorithms that are not sufficiently regularized. Similarly, Lyu & Li (2019) show theoretically and experimantally that the the details of the training procedure can significantly change the robustness of CNNs.

In this paper we give an alternative perspective on adversarial examples by focusing on *Bayes-Optimal* classifiers. We ask: under what conditions on the data distribution will a large fraction of the points be close to the *optimal* decision boundary? We show that it is easy to construct data distributions in which this happens (bottom of figure 2) and seemingly similar distributions in which this does not happen (top of figure 2). Contrary to previous explanations, adversarial attacks may succeed when the optimal decision surface is continuous and the data does not need to lie in high dimensions. Rather we show that the susceptibility to adversarial examples is related to the presence of asymmetries in the predictive power of different features in the data distribution, and we show analytically that when such asymmetries exist, optimal classifiers will suffer from adversarial examples. We introduce new datasets of realistic images for which the Bayes-Optimal classifier can be calculated efficiently and show that for some of these datasets the optimal classifier is robust to adversarial examples and in others it is not.[1] In systematic experiments with datasets where the optimal classifier is robust, we show that standard CNN training methods consistently find a non-robust classifier even though the optimal classifier is robust, while large-margin methods often find a robust classifier using exactly the same training data. Our results suggest that adversarial vulnerability is not an unavoidable consequence of machine learning in high dimensions, and in many cases may be a result of suboptimal training methods.

---

[1]The datasets and models will be made publicly available after publication.

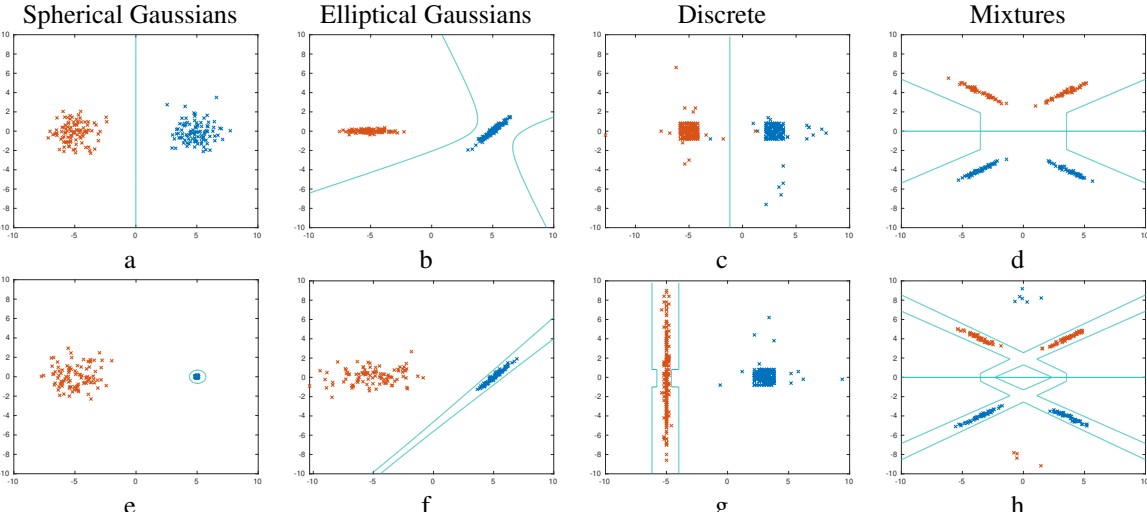

Figure 2: Top: examples of 2D distribution where the optimal decision boundary is far from the datapoints. Bottom: distribution where the optimal boundary is close to many of the datapoints.

## 2 ANALYSIS: IS THE OPTIMAL CLASSIFIER NECESSARILY ROBUST?

We focus on a two class classification problem and denote by $p_1(x)$ the distribution of $x$ when $x$ is in the first class, and similarly $p_2(x)$ for the second class. We assume that $p_1(x), p_2(x)$ are known and that the two classes have equal priors, so the *Bayes-Optimal* classifier simply classifies $x$ as belonging to class 1 if $p_1(x) > p_2(x)$ and to class 2 otherwise. It is well known that this classification rule is optimal and no other classification rule can achieve higher accuracy (assuming of course that $p_1(x), p_2(x)$ are correct) (Duda et al., 1973). A textbook example of Bayes-Optimal classification is when both classes are generated with a Gaussian with the same spherical covariance matrix, in which case the optimal classifier is a linear discriminant which is orthogonal to the difference between the two means (figure 2a). In this case, if the distance between the two means is large relative to the covariance, then almost all points are far from the decision boundary and so an adversarial attack which only makes small changes to the input will typically fail. But as shown in the bottom of figure 2 there are other examples where the decision boundary is close to many of the datapoints and an adversarial attack which only makes small changes to the input will often succeed. What distinguishes these two cases?

To gain intuition, consider the images shown in the left side of figure 3. Assume we have infinite examples of images of Cindy Crawford and denote by $p_1(x)$ the probability of generating an image under this distribution. Denote by $p_2(x)$ the probability of generating an image of a different person. Under $p_1(x)$ all images have a mole, while under $p_2(x)$ this probability is very low. Thus an optimal classifier will assign an extremely high weight to the presence of a mole in an image. This means that by making a tiny change in the image and erasing the mole (right side of figure 3) we can drastically change the output of the optimal classifier (and in fact drive $p_1(x)$ down to zero). Note that this is *not* the same as the standard definition of overfitting (i.e. a difference between performance on the training samples and test samples): even unseen images of Cindy Crawford that are generated from $p_1(x)$ will have a mole present, so giving this feature high weight will not hurt generalization, but will make the classifier susceptible to adversarial attacks. This intuition is similar to that used in the recent work of (Ilyas et al., 2019) who argue that adversarial vulnerability is a feature, not a bug, and results from the presence of predictive features that are not robust. The way we formalize this intuition, below, is quite different from that of (Ilyas et al., 2019) (whose analysis focused on linear classifiers) and also leads to quite different conclusions.

We first focus on the case where both class distributions are Gaussians, i.e. $p_1(x), p_2(x)$ are both Gaussian with means $\mu_1, \mu_2$ and covariances $\Sigma_1, \Sigma_2$.

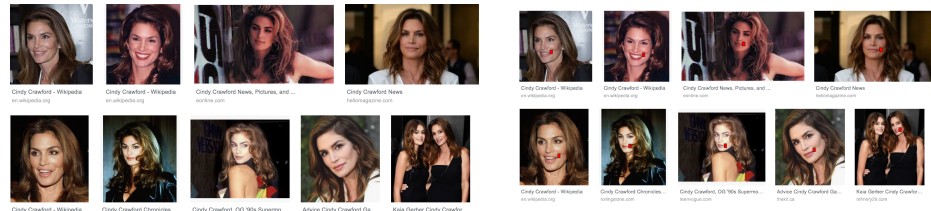

Figure 3: Intuition behind our analysis. When the distribution of one of the classes includes an extremely predictive feature (e.g. the mole in images of Cindy Crawford), then the optimal classifier assigns this feature an extremely large weight. This means that by changing just that one feature, an adversary can drastically lower the probability that it be classified correctly. In the text we formalize this intuition.

**Observation 1:** Let $d$ be a direction of minimal variance under $\Sigma_1 : d = \arg\min d'^T \Sigma_1 d'$. Let $\sigma_1^2$ be the variance projecting $x$ onto that direction when $x$ comes from class 1. If $\sigma_1 \to 0$ and $\Sigma_2$ is full rank then almost any point in class 1 is arbitrarily close to the optimal decision surface.

**Proof:** We denote by $\sigma_2 = d^T \Sigma_2 d$ the variance of the data in direction $d$ under the distribution of the second class $p_2$. Note that by the assumption that $\Sigma_2$ is full rank, this variance must be nonzero. We write the vector $x$ as $(t, s)$ where $t$ is the projection in direction $d$ and $s$ is a vector of projections in directions orthogonal to $d$. We denote by $t_i$ the projection of $\mu_i$ in direction $d$. The decision surface as a function of $t$ is a solution to:

$$\left(\frac{1}{\sigma_1^2} - \frac{1}{\sigma_2^2}\right) t^2 + \left(2\frac{t_2}{\sigma_2^2} - 2\frac{t_1}{\sigma_1^2}\right) t + \left(\frac{t_1^2}{\sigma_1^2} - \frac{t_2^2}{\sigma_2^2} + \log \frac{\sigma_1^2}{\sigma_2^2}\right) = \log p_2(s|t) - \log p_1(s|t) \quad (1)$$

Now since $d$ is a direction of minimal variance it must be an eigenvector of $\Sigma_1$ so that $s$ and $t$ are independent under $p_1$ and we can write $\log p_1(s|t) = \log p_1(s)$. Using the standard equation for conditional Gaussians we see that $p_2(s|t)$ is a Gaussian with the following means and variances.

$$\mu_{s|t} = \mu_s + \Sigma_2^{st}(t - t_2) \quad (2)$$

$$\Sigma_{s|t} = \Sigma_2^{ss} - \Sigma_2^{st}\frac{1}{\sigma_2^2}\Sigma_2^{ts} \quad (3)$$

where $\Sigma_2^{ss}, \Sigma_2^{st}$ are the appropriate submatrices of the covariance matrix $\Sigma_2$.

As $\sigma_1 \to 0$ then $t \to t_1$ so we can approximate $\log p_2(s|t)$ with a Gaussian with the following means and variances:

$$\mu_{s|t} = \mu_s + \Sigma_2^{st}(t_1 - t_2) \quad (4)$$

$$\Sigma_{s|t} = \Sigma_2^{ss} - \Sigma_2^{st}\frac{1}{\sigma_2^2}\Sigma_2^{ts} \quad (5)$$

Note that now neither the mean nor the variances depend on $t$. This means that the decision surface for $t$ is a solution to the following equation:

$$\left(\frac{1}{\sigma_1^2} - \frac{1}{\sigma_2^2}\right) t^2 + \left(2\frac{t_2}{\sigma_2^2} - 2\frac{t_1}{\sigma_1^2}\right) t + \left(\frac{t_1^2}{\sigma_1^2} - \frac{t_2^2}{\sigma_2^2} + \log \frac{\sigma_1^2}{\sigma_2^2}\right) = C(s) \quad (6)$$

where the right hand side is a constant that does not depend on $t$ or $\sigma_1$. As $\frac{\sigma_1}{\sigma_2}$ approaches zero, the solutions of this equation approach $t_1$. And because $\sigma_1 \to 0$ almost all samples from the first Gaussian will be close to $t_1$ so moving $x$ by a tiny amount in direction $d$ will change the optimal decision. $\square$

As can be seen from equation 6, we do not need $\sigma_1$ to be exactly equal to zero for the decision surface to be close to the datapoints. It is enough for $\sigma_1$ to be significantly smaller than the minimal variance of $\Sigma_2$. Figures 2a,b,e,f illustrate this dependence: when the minimal variances in both Gaussians is similar then the decision boundary is far from most points (top), but when there is a strong asymmetry in the minimal variances then the decision boundary becomes close to the datapoints in one of the classes. In (Tanay & Griffin, 2016), the existence of directions of small variance

was suggested as a possible explanation for adversarial examples by invoking overfitting: when such directions exist, a linear classifier may fail to learn the optimal decision boundary. Our analysis, on the other hand, is for the Bayes-Optimal classifier which by definition does not overfit.

We now assume that the distribution in each class can be represented as a mixture model:

$$p_i(x) = \sum_k \pi_{ik} p_{ik}(x) \tag{7}$$

where $\pi_{ik}$ represent the prior probability of component $k$ in $p_i$ and $p_{ik}$ is the probability distribution of that component.

We also assume that within each class, the components are well separated, i.e. that for each datapoint the assignment probabilities put all their mass on one of the components. More formally, denote by $g_k(x)$ the assignment probability of a datapoint $x$ to component $k$:

$$g_{ik}(x) = \frac{\pi_{ik} p_{ik}(x)}{\sum_j \pi_{ij} p_{ij}(x)} \tag{8}$$

then we assume that for each $x$, $g_{ik}(x) = \delta(k - k_i^*(x))$ where $k_i^*(x)$ is the index of the component that is most likely to have generated $x$ under probability $p_i(x)$. Under this assumption the probability of generating a point $x$ under $p_1$ is simply:

$$\pi_{k_1^*(x)} p_{1k_1^*(x)}(x) \tag{9}$$

We will also assume that for well separated components, within each distribution, the assigned component does not change when we perturb $x$ by a small amount $k_i^*(x) = k_i^*(x + \epsilon)$ where $\epsilon$ is a small perturbation.

**Observation 2:** Assume that both classes are generated by well separated Gaussian mixtures. Let $\Sigma_{1k}$ be the covariance matrix of the $k$th Gaussian in the mixture for class 1. If there exists a Gaussian $j$ in class 2 such that $\Sigma_{1k}$ and $\Sigma_{2j}$ satisfy the asymmetry conditions of observation 1, then almost any point generated by that Gaussian is close to the optimal decision boundary.

**Proof:** We simply apply observation 1 to the two Gaussians, $\Sigma_{1k}$ in class 1 and $\Sigma_{2j}$ in class 2. By observation 1, we can make all points generated by the Gaussian in class 1 more likely to have been generated by the Gaussian in class 2, by moving a tiny amount. By the well separatedness asssumption, the original Gaussian in class 1 is still more probable than all the other Gaussians in the same class, so the point will now be classified as coming from class 2 by the Bayes-Optimal classifier. □

Figures 2d,h illustrate the importance of asymmetry in data generated from a Gaussian Mixture Model. In the top figure, both classes are generated from a mixture of two Gaussians, whose minimal variance in the same. In the bottom figure, each class data is generated by a mixture of three Gaussians: the two original Gaussian plus a third isotropic Gaussian that has low mixing proportion but very large variance. Once this third Gaussian is present in the other class, the Bayes-Optimal decision surface becomes very close to all points in both classes (except for those generated by the rare, broad Gaussian).

## 2.1 Non Gaussian distributions

A natural question following observation 1 is to what extent the result depends on the Gaussian distribution. To address this, we now consider *discrete* distributions. We assume that every instance $x$ is described by quantized features that can take on a discrete number of values. For example, the features can be wavelet coefficients of an image that are discretized into 256 possible values. This means that $p_1(f), p_2(f)$ are simply very large tables that give the probability of observing a particular discrete set of image features given each of the classes. Of course learning such a large table is infeasible without additional assumptions, but recall that we are analyzing the Bayes-Optimal case, where we assume $p_1(f), p_2(f)$ are known.

**Observation 3:** Assume there exists a feature $i$ and a quantization level $k$ so that $p_1(f_i = k) \to 1$. Assume also that for any feature vector $f$, $p_2(f) > 0$. Then almost any point in class 1 is one quantization level away from the optimal decision boundary.

**Proof:** We again write $f = (s, t)$ where $s$ is the $ith$ feature and $t$ are all other features.

$$p_1(s, t) = p_1(s)p_1(t|s); \tag{10}$$

Now assume that $p_1(s)$ approaches 1 for $s = k$ and 0 otherwise. This means that for almost any point in class 1 the value of that feature is equal to $k$. We now change the feature by one quantization level and obtain a new feature vector $(\tilde{s}, t)$ and $p_1(\tilde{s}, t) = p_1(\tilde{s})p_1(t|\tilde{s}) = 0$. On the other hand, by the assumption $p_2(\tilde{s}, t) > 0$ so that this point would now be classified as belonging to class 2. $\square$

As in the Gaussian case, we do not need $p_1(s)$ to be exactly equal to a delta function for the decision surface to be close to most points. It is enough that $\frac{p_1(\tilde{s})}{p_1(\tilde{s})}$ be much smaller than the minimal values of $p_2$ for the decision to be flipped when we replace $s$ with $\tilde{s}$. Figures 2g,c illustrate this dependence. In both cases, the data is sampled from a discrete distribution where the features are simply discretization of the two spatial coordinates into 100 levels each. In other words, $p_1$ and $p_2$ are tables of size $10,000$ and each entry in the table represents the probability of generating a point at one of the $10,000$ possible locations. In the top example, the minimal value of the probability table is approximately the same in both classes, while in the bottom example, there is a strong asymmetry and the decision boundary becomes close to all points in one of the classes.

**Observation 4:** Assume that both classes are generated by well separated discrete feature distributions. Let $p_{1k}(f)$ be be the $k$th table of feature distributions under class 1. If there exists a $j$ in class 2 such that $p_{1k}(f), p_{2j}(f)$ satisfy the asymmetry conditions of observation 2, then almost any point generated by that component is close to the optimal decision boundary.

The proof is identical to the proof of observation 2.

## 3 DATASETS OF REALISTIC IMAGES WITH EFFICIENT CALCULATION OF THE BAYES-OPTIMAL CLASSIFIER

In order to connect the discussion of optimal classifiers with datsets that can be used with modern classifiers, we introduce new datasets of realistic images of faces and digits for which the Bayes-Optimal classifier can be calculated efficiently. We do this by leveraging recent advances in generative modeling of full images (Goodfellow, 2016; Richardson & Weiss, 2018).

To create these datasets, we start with a labeled training set (e.g. MNIST) with two classes. We then train a separate Mixture of Factor Analyzers (MFA) model $p_1(x)$, $p_2(x)$ on images from the two classes. The MFA model (Ghahramani et al., 1996) is a standard density model: it models each density as a mixture of Gaussians where the covariance of each Gaussian is the sum of a low rank matrix and a diagonal one (see appendix for details). We now create a new training set by sampling images from the two models, and similarly a new test set. Since these datasets were created by sampling from a known model, we can calculate the Bayes-Optimal classifier. At the same time, the images are realistic and MFA models have been shown to capture much of the variability of the images in the original data (Richardson & Weiss, 2018). The top of figure 4 shows samples from the $p_1(x), p_2(x)$ when the MFA were learned separately from the "male" and "female" attributes in the CelebA dataset (Liu et al., 2015). More samples from other datasets including digits and faces can be found in the appendix. While these samples are typically somewhat blurred, it can be seen they are realistic and highly variable. In fact, in many real world applications one needs to classify somewhat blurry images (e.g. analyzing faces in surveillance videos).

In order to investigate the importance of asymmetry in the variances, we created symmetric and asymmetric variants of these datasets. In the symmetric version, we regularized the MFA so that all covariance matrices had the same minimal variance. In the asymmetric version, we added to each MFA model one additional "outlier" component: it had a diagonal covariance with much larger minimal variance than all the other covariances and a mean which is close to the mean of the data.

The advantage of using a MFA model over other generative models such as VAEs or GANs (Kingma & Welling, 2014; Gulrajani et al., 2017) is that the log likelihood of any image can be calculated efficiently. Specifically we follow (Richardson & Weiss, 2018) and find the Gaussian component that is most likely to have generated the image, and then evaluate the likelihood of the image given that Gaussian. We can then use this log likelihood to classify an image as belonging to one of the two classes (e.g. "male" or "female"). Since the data were generated by the assumed distributions,

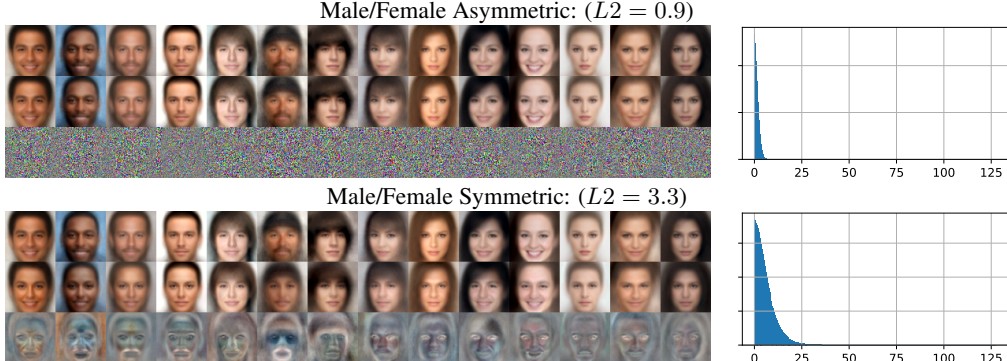

Figure 4: We introduce new datasets of realistic images for which the Bayes-Optimal classifier can be calculated efficiently. The vulnerability of the optimal classifier depends on the presence of asymmetries. Each result shows the original images, adversarial images and perturbations (magnified for visibility as necessary). Histogram of the perturbations in pixel values (0-255) are on the right.

this classifier is Bayes-Optimal. Indeed in all datasets we created, the accuracy of the classifier was close to 100%. We now asked: is this Bayes-Optimal classifier robust to adversarial attacks?

Unlike our theoretical results, deciding whether or not a real classifier is robust or not requires an operational definition of what constitutes a "tiny" or "imperceptible" perturbation. We follow the standard practice of calculating the mean L2 perturbation of an adversarial attack. In other words, we allow the adversary an unlimited budget in attacking the classifiers, and we then measure how large (in terms of Euclidean norm) a perturbation was required to cross the decision boundary. The mean is calculated only over successful attacks, when the original sample was correctly classified and the adversarial example was not. Since this definition is sensitive to outliers and the particular choice of Euclidean norm, we also examined the histograms of changes made to each pixel in the adversarial attack. Finally, we visually inspected the adversarial images.

In all 15 datasets, we found that these three methods of defining robustness are consistent. For the face images, when the mean L2 is less than $1.5$, then the adversarial images are almost indistinguishable from the original images, and the vast majority of the pixels in the adversarial images are within $5/255$ intensity levels from their original value. On the other hand, when the mean L2 is around 3, then the adversarial images are perceptually quite different from the original ones, and many pixels differ by more than $5/255$ from their original values. Similar definitions of robustness were used in previous reports (Schott et al., 2019; Carlini et al., 2019).

We used a simple gradient attack in which we take a small step in the direction of the gradient of the MFA log likelihood (see appendix for details). Similar results are achieved with a standard implementation (Papernot et al., 2016) of the *CW-L2* attack (Carlini & Wagner, 2017).

As shown in figure 4 the difference between the symmetric datasets and asymmetric datasets is dramatic (see appendix for similar results on other classes). When there exists a large asymmetry between the minimal variances of different Gaussians, the conditions of observation 3 hold, and a tiny imperceptible change is sufficient to fool the Bayes-Optimal classifier. However, when all Gaussians have the same minimal variance, the adversary needs to make much larger changes and the adversarial examples become perceptually meaningful.

# 4    EXPERIMENTS: WHY ARE CNNS SO BRITTLE?

Given our analysis, the fact that modern machine learning methods are often susceptible to tiny adversarial attacks may be due to two very different reasons (figure 5). One reason could be that the data distribution is asymmetric, so that the Bayes-Optimal classifier is not robust, and hence it is not surprising that a CNN is also not robust. A second possible reason is illustrated in figure 5b: here the data distribution is symmetric and the Bayes-Optimal classifier is robust, yet SGD starting from a bad initial condition finds a brittle classifier. If this is the case, then the brittleness is not due to the data distribution but rather a failure of the learning method. Indeed (Tanay & Griffin,

Asymmetric distributions        Non-Optimal Learning

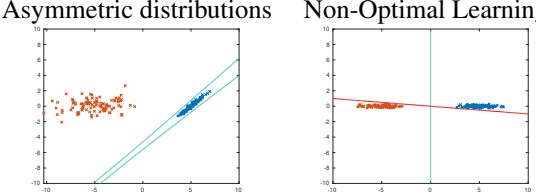

Figure 5: Why are CNNs so brittle? Is it because the data distributions are asymmetric so that the optimal classifier is also brittle (left) or is due to non optimal learning in cases where the optimal classifier is robust (right)?

2016; Nakkiran, 2019) have presented toy distributions where the optimal classifier is robust but suboptimal learning fails to find it.

In order to separate the contribution of the dataset from the estimation method in the vulnerability of machine learning methods, we trained a CNN on samples from all 15 "symmetric" datasets described in section 3, and measured the vulnerability on the learned CNN. We used the standard CNN implementation available as part of the *CleverHans* library. We asked: will the CNN find a brittle classifier even though the optimal one is robust?

Results are shown in figure 6. *In all 15 cases, the CNN found a high accuracy classifier that was vulnerable to small adversarial attacks, even though the optimal classifier is robust.* The difference is most dramatic in the celebA tasks, where the CNN adversarial examples are almost indistinguishable from the original images (examples of the attacks on different datasets are shown in the appendix). In the easier, MNIST tasks, the CNN adversarial perturbations are still significantly smaller than those of the Bayes-Optimal classifier and less perceptually meaningful. While there are many possible architectures and optimization methods for CNNs, we did not find any improvement in the CNN robustness in our attempts to change the number of filters, layers, training iterations etc. In particular, (Schmidt et al., 2018) have argued that one needs more training examples to achieve robust classification, so we systematically varied the amount of training images (generated dynamically at each SGD iteration), and found no significant improvement in robustness as we increased the number of training examples up to 1 million examples (see appendix for details). Even adversarial training of the CNN (using the method of (Zhang et al., 2019)) did not significantly improve robustness.

Is it possible to learn a robust classifier for these datasets from finite training data? To answer that question, we then trained linear and RBF support vector machine classifiers on exactly the same datasets. The linear SVM attempts to maximize the margin while maintaining high accuracy, but since the optimal classifier is nonlinear it ends up learning a brittle classifier. More importantly, *with an appropriate bandwidth parameter RBF SVMs often find a robust classifier when trained on exactly the same data* (when the bandwidth parameter is too large, the RBF performs similarly to a linear SVM). Returning to figure 5, our results strongly support the hypothesis that for these cases brittleness is due to suboptimal learning methods, even when the data distributions are symmetric and the optimal classifier is robust.

## 5 DISCUSSION

Since the initial discovery of adversarial examples for CNNs there has been much discussion whether they are a "bug" that is specific to neural networks or a "feature" of high dimensional geometry. On the one hand, our results show that even the Bayes-Optimal classifier may be susceptible to tiny adversarial perturbations, and this can happen in low dimensions and when the optimal classification function is smooth. Perhaps more significantly, our analysis has also enabled us to construct realistic datasets for which the Bayes-Optimal classifier is robust. We find that standard CNN training consistently fail to find a robust classifier for these datasets, while large-margin methods can succeed when training on exactly the same data. This suggests that in some situations, the presence of adversarial examples represents a failure of current, suboptimal, learning methods, rather than being an unavoidable property of learning in high dimensions.

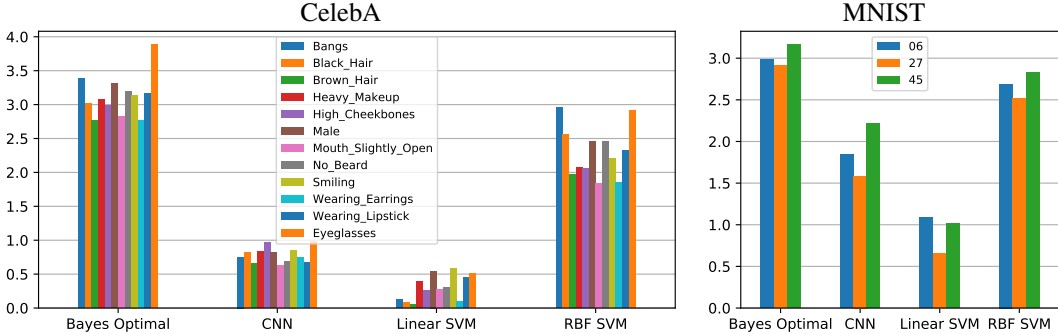

Figure 6: Adversarial attack perturbation sizes (mean L2 norm) for different models for all symmetric datasets.

We are by no means advocating a return to using RBF SVMs rather than CNNs. Rather, we believe that explicit regularization methods for CNNs may enable learning robust classifiers while maintaining the power of deep architectures. Recent theoretical work on gradient descent methods suggests that they implicitly reward large margin classifiers in both shallow and deep architectures (Soudry et al., 2018; Poggio et al., 2017; Lyu & Li, 2019) although convergence to a large margin classifier may require exponential time.

While our experiments have focused on the "symmetric" datasets, where there is no tradeoff between accuracy and robustness, there is no reason to believe that real datasets will be symmetric rather than asymmetric. As our analysis has shown, if the real datasets do have strong asymmetries, then the Bayes-Optimal classifier will not be robust. Nevertheless, improved learning algorithms should allow us to trade off robustness and accuracy in a principled manner, as can currently be done in the training of shallow architectures such as SVMs. In the appendix, we show results with trained models on real data for the CelebA Male/Female task. Although we cannot calculate the Bayes Optimal classifier, the trend is similar to what we saw in the synthetic data: CNNs and Linear SVMs learn very vulnerable models, while the RBF SVM learns a classifier that is robust and can only be fooled when the adversary makes perceptually meaningful changes.

In general, when trying to understand a complex effect, it is often useful to disentangle the different causes. The Bayes-Optimal perspective on adversarial examples identifies two possible causes: asymmmetries in the datasets and suboptimal learning. Furthermore, it allows us to disentangle these two possible causes by creating synthetic datasets in which one of the two causes can be clearly implicated. We are optimistic that such an approach will be of great use in developing new learning algorithms that are practical and robust.

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

## A  SYMMETRIC VS. ASYMMETRIC

Figure 7 presents additional examples comparing symmetric and asymmetric datasets and the relative robustness of their Bayes-Optimal classifiers to adversarial examples.

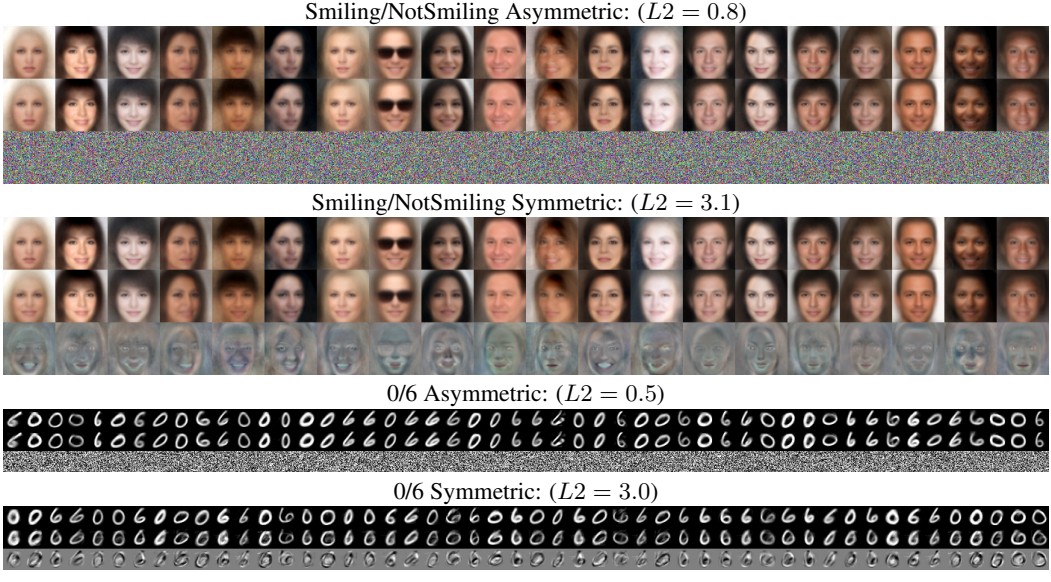

Figure 7: The vulnerability of the optimal classifier depends on the presence of asymmetries. Each result shows the original images, adversarial images and perturbations (magnified for visibility as necessary). In the symmetric datasets, attacking the optimal classifier requires large and perceptually meanigfull perturbations.

## B  MODELS

In this section we provide additional information about the different classification models – architecture, hyper-parameters and training procedure.

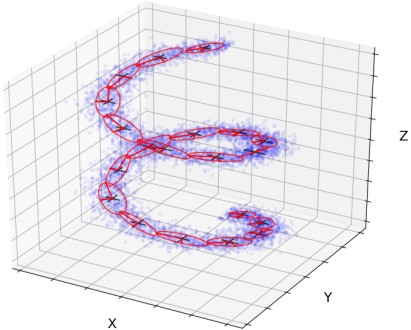

Figure 8: A Mixture of Factor Analyzers (MFA) model for toy data in $\mathbb{R}^3$ (sampled 3D helix – blue points). The latent dimension is 2 – each component is a Gaussian on a learned 2-dimensional hyperplane with added axis-aligned noise.

### B.1 MFA

A Mixture of Factor Analyzers (MFA) (Ghahramani et al., 1996) is a Gaussian Mixture Model where each component is a Factor Analyzer parameterized by a *low rank plus diagonal* covariance matrix. MFA provides a good tradeoff between the non-expressive diagonal-covariance model and a full-covariance model, which is too computationally expensive for high-dimensional data such as full images.

The model for a single Factor Analyzer component is:

$$x = Az + \mu + \epsilon \,, z \sim \mathcal{N}(0,\, I) \,, \epsilon \sim \mathcal{N}(0,\, D) \,, \tag{11}$$

where $A$ is the rectangular *factor loading* matrix, $z$ is a low-dimensional latent factors vector, $\mu$ is the mean and $\epsilon$ is the added noise with a diagonal covariance $D$. This results in the Gaussian distribution $x \sim \mathcal{N}(\mu,\, AA^T + D)$. The MFA is a mixture of such Gaussians. Figure 8 demonstrates the MFA model for toy data.

The MFA model was trained using the code provided by Richardson & Weiss (2018). The models are trained using Stochastic Gradient Descent. The training data (CelebA, MNIST) is first split by the desired binary attribute (e.g. Smiling / Not Smiling) and then a separate MFA model was trained independently for each subset of training samples. Because of imbalance in the number of samples per class in CelebA, we set the number of components as the number of samples divided by 1000. For MNIST we used a fixed value of 25 components per class. We chose an MFA latent dimension of 10 for CelebA and 6 for MNIST.

To allow attacking the MFA model with standard adversarial attacks such as CW-L2, we implemented it in TensorFlow as a standard CleverHans (Papernot et al., 2016) model.

### B.2 BAYES-OPTIMAL

The MFA model is the Bayes-Optimal classifier when the data is sampled from the model. We modified the trained MFA models to define pairs of Bayes-optimal models.

A single Factor Analyzer component is defined by a mean vector $\mu$, a rectangular matrix $A$ and per-pixel noise variance $D$ (diagonal). The component distribution is then $x \sim \mathcal{N}(\mu,\, AA^T + D)$.

For the *symmetric* models, we simply fixed all noise variance values in $D$ to a small value (std of 0.001). For the *asymmetric* case we added two outlier components (one for each class) that are modifications from the dataset global mean in directions of low-variance: We performed PCA over the entire dataset and took the eigenvector for the 50th largest eigenvalue as this direction, where the mean of one outlier is in the positive direction and the mean of the other in the negative one. We set $A = 0$ and $D = 0.5$ for both outliers, making them spherical gaussians around the two means.

### B.3 CNN

We used the reference fully-convolutional network implemented as part of the CleverHans library (*ModelAllConvolutional*). The architecture consists of 2D convolution layers with a kernel size of 3 followed by *Leaky ReLU* activations and average pooling. We set the basic number of filters (nb_filters) to the minimal required number that achieves 100% classification accuracy on our datasets. All other hyper-parameters were left at their default values (calculated automatically by the library according to the image resolution). Optimization method is *Adam*. The reported results are not sensitive to the specific architecture (e.g. number of layers, channel depth) and training parameters.

### B.4 LINEAR SVM

We used the standard 2-class linear SVC implementation provided by sklearn (no hyper-parameters). Linear SVM is trained directly on the vectorized image samples. The learned model consists of a weight vector $W$ and a scalar bias $b$. The decision for a sample $x$ is simply $\text{sign}(W^T x + b)$.

### B.5 RBF SVM

We used sklearn for the Radial Basis Function (RBF) kernel SVM as well. A single hyper-parameter is required, $\gamma = \frac{1}{2\sigma^2}$. We chose the highest $\gamma$ value that still provides a high classification accuracy.

## C ATTACKS

### C.1 CW-L2

The Carlini & Wagner L2 attack (Carlini & Wagner, 2017) is a recommended strong attack that minimizes the perturbation L2 norm. The attack minimizes a weighted combination of a classification loss with the perturbation L2 size. The relative weight is a parameter that is found using a binary-search. We used the CleverHans implementation with the following hyper-parameters: 500 iterations, 3 binary-searches and a learning rate of 0.01.

### C.2 LD-GRAD

Since the MFA and SVM models provides a simple closed-form expression for the likelihood and its gradient, we implemented a simple and fast version of a gradient-attack for these models. Our attack performs multiple (typically 3) steps in the direction of the gradient of the difference in log-likelihood between the source and target Gaussian components. We compute the required step size to cross the gap in the log-likelihood between the source and the target components. We repeated some of the experiments with the (much slower) CW-L2 attack and verified that the results are similar (i.e. models that are shown to be robust to the LD-Grad attack are also robust to the CW-L2 attack with similar perturbation magnitudes).

## D ADDITIONAL RESULTS

In this section we provide additional experimental results for the different datasets, attributes and models. Table 1 lists the clean and adversarial classification accuracy of all models for all datasets and attributes.

### D.1 CNN – LONGER TRAINING AND WIDER NETWORKS

As shown in Figure 6, unlike the Bayes-optimal and the SVM RBF classifiers, CNNs trained on the symmetric datasets are vulnerable to adversarial examples (required perturbation size is small). Here we investigate the effect of longer training with additional training samples and of wider networks.

We trained a CNN with 8 convolution layers and tested different values for the width (number of output channels) of the first layer. Each second layer has a stride of 2 and the width is doubled, so

| Dataset / Attribute | Bayes-Optimal | CNN | Linear SVM | RBF SVM |
|---|---|---|---|---|
| CelebA / Bangs | 100% (0%) | 100% (0%) | 100% (0%) | 96% (18%) |
| CelebA / Black Hair | 100% (0%) | 98% (2%) | 100% (0%) | 96% (26%) |
| CelebA / Brown Hair | 100% (0%) | 99% (1%) | 100% (0%) | 86% (16%) |
| CelebA / Heavy Makeup | 100% (0%) | 100% (0%) | 100% (0%) | 96% (14%) |
| CelebA / High Cheekbones | 100% (0%) | 100% (0%) | 100% (0%) | 96% (10%) |
| CelebA / Male | 100% (0%) | 100% (0%) | 100% (0%) | 100% (8%) |
| CelebA / Mouth Slightly Open | 100% (0%) | 100% (0%) | 100% (0%) | 94% (8%) |
| CelebA / No Beard | 100% (0%) | 100% (0%) | 100% (0%) | 94% (10%) |
| CelebA / Smiling | 100% (0%) | 100% (0%) | 100% (0%) | 96% (10%) |
| CelebA / Wearing Earrings | 100% (0%) | 99% (1%) | 100% (0%) | 94% (16%) |
| CelebA / Wearing Lipstick | 100% (0%) | 100% (0%) | 100% (0%) | 92% (18%) |
| CelebA / Eyeglasses | 100% (0%) | 100% (0%) | 100% (0%) | 100% (22%) |
| MNIST / 06 | 100% (0%) | 100% (0%) | 100% (0%) | 100% (4%) |
| MNIST / 27 | 100% (0%) | 100% (0%) | 100% (0%) | 100% (7%) |
| MNIST / 45 | 100% (0%) | 100% (0%) | 100% (0%) | 100% (8%) |

Table 1: Classification accuracy values for the different symmetric datasets for the original and adversarial (in brackets) samples.

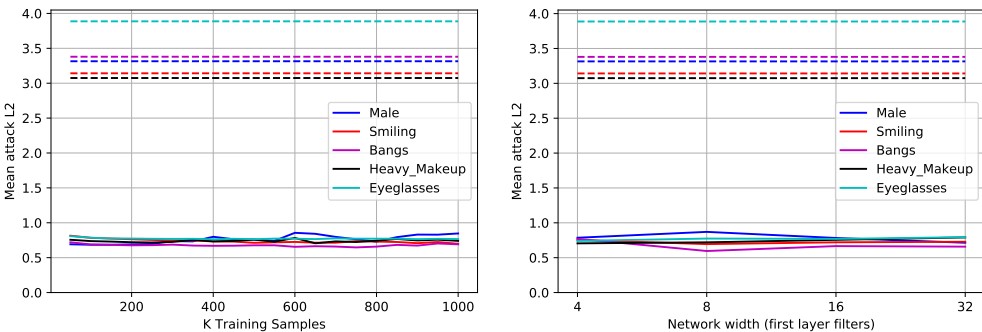

Figure 9: Training wider CNNs with more samples does not improve the adversarial robustness. The Bayes-optimal classifiers are shown as dashed lines.

the width of the final layer was 16 times that of the first layer. To test the effect of longer training with more samples, we modified the training procedure so that new samples from the symmetric dataset were generated dynamically. We trained the networks for a total of 1 million samples, in mini-batches of 50. The results are shown in Figure 9, indicating that wider networks and more data does not improve robustness.

## D.2 INCREASING THE DATA NOISE VARIANCE

As described in Appendix B.1, in the main experiments we fixed the noise standard deviation (the added diagonal part of the MFA covariance) of all Gaussians in the model that generated the symmetric dataset to $\sigma = 0.001$. Here we verify that our results are not sensitive to this value. We increased the noise std to 50 times the original value. At these levels (Figure 10, left) the noise is clearly visible in the data samples. As can be seen (Figure 10, right), the effect on the robustness of trained CNN models is minor. As expected, the Bayes-Optimal model does not change its decision boundary and remains robust.

## D.3 ROBUST (ADVERSARIAL) CNN TRAINING

To test the effect of adversarial training on the CNN model for the symmetric dataset, we used the method proposed by Zhang et al. (2019), which won the NeurIPS 2018 Adversarial Vision Challenge (Robust Model Track). Although slightly more robust, the robustified CNN is still far from the Bayes-optimal model (Figure 11).

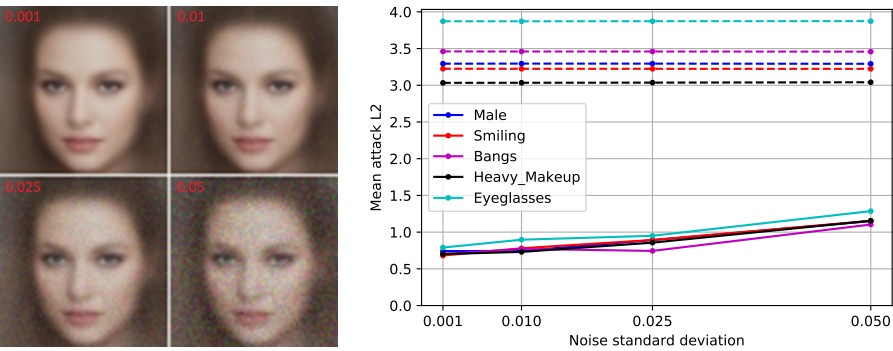

Figure 10: Increasing the noise standard deviation by a factor of 50 has no effect on the robustness of the Bayes-Optimal models (dashed lines) and only a small effect on the robustness CNNs trained on the noisy samples.

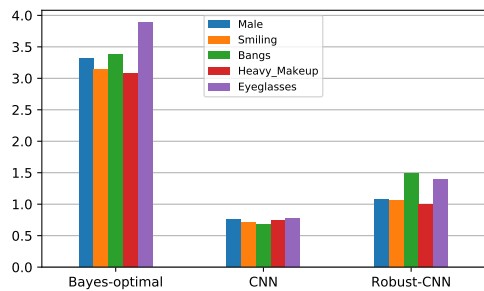

Figure 11: The effect of robust (adversarial) training per Zhang et al. (2019) with $\beta = 10$ (favouring robustness), shown together with the Bayes-optimal for reference.

### D.4   RESULTS ON REAL DATA

#### D.4.1   TRAINING ON SYMMETRIC DATA AND TESTING ON REAL DATA

To estimate how close our symmetric datasets are to the real datasets, we tested the CNNs that were trained on the symmetric dataset on real test samples and compared the test accuracy to that of CNNs that were trained on the real training data. As can be seen in Table 2, there is an average accuracy reduction of just 5%, indicating that the symmetric datasets are not that far from the original data.

#### D.4.2   TRAINING AND TESTING ON REAL DATA

Our experiments on the symmetric data (Figure 6) showed that, on all tested datasets, SVM with RBF kernel manage to find a decision boundary that is almost as robust as the optimal one, while CNNs and Linear SVMs find (probably due to two different reasons) a vulnerable boundary, which can be attacked with a much smaller perturbation size.

One can ask, do these results represent the real data? To answer this questions we trained the three models on one of the real datasets – CelebA Male/Female and indeed, as shown in Figure 12 the

| Dataset / Attribute | Real Train Data | Symmetric Train Data |
|---|---|---|
| CelebA / Male | 92.3% | 88% |
| CelebA / Smiling | 89.5% | 85.6% |
| CelebA / Bangs | 94.3% | 90.4% |
| CelebA / Heavy Makeup | 86.3% | 79.9% |
| CelebA / Eyeglasses | 98.4% | 91.7% |

Table 2: Real test images classification accuracy for CNNs trained on real training data vs CNNs trained on samples from the symmetric dataset. Accuracy shown is for the robust CNNs.

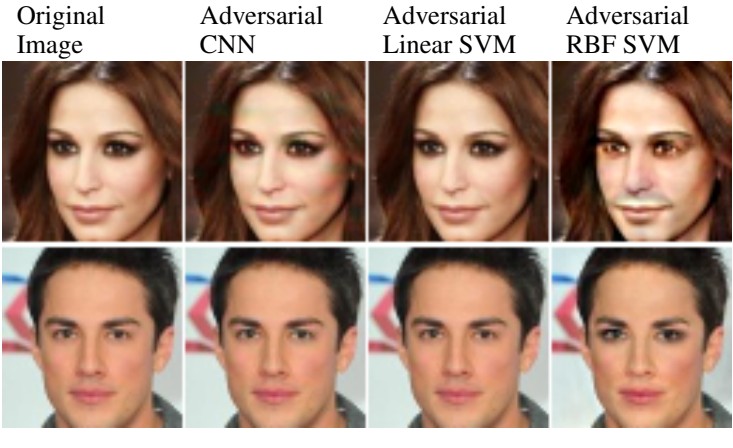

| Model: | CNN | Lin. SVM | RBF. SVM |
|---|---|---|---|
| Clean accuracy | 96.3% | 90.0% | 89.3% |
| Adversarial accuracy | 3.7% | 10% | 16% |
| Mean attack L2 | 0.59 | 0.15 | 2.99 |

Figure 12: Results of trained models and adversarial attacks on real data.

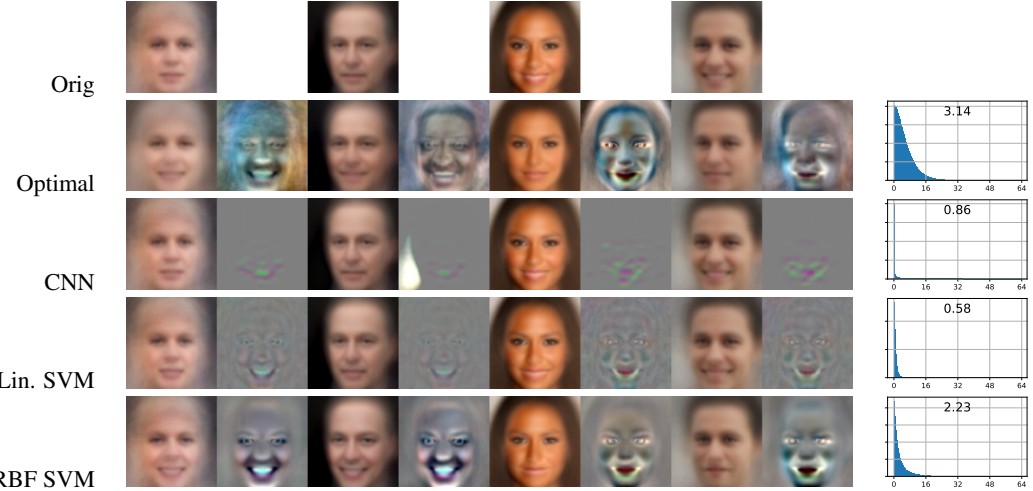

Figure 13: Samples, perturbations and histograms for CelebA attribute 'Smiling'

results are similar to the synthetic symmetric dataset – CNN and Linear SVM are vulnerable and RBF SVM is robust.

Note that unlike our proposed symmetric data, in which the "gold standard" optimal classifier is both robust and accurate, on real data, which may contains variance asymmetries, different models might reach different trade-off points between accuracy and robustness. In particular, it is known that RBF SVM accuracy is highly influenced by the hyperparameters $C, \gamma$ and we performed only a minimal search to obtain these results. In future work, it would be interesting to explore the full regularization path as in (Hastie et al., 2004).

### D.5 ADDITIONAL RESULTS FOR THE SYMMETRIC DATASET

Figures 13-18 show original and adversarial samples and perturbations as well as histograms of the perturbations in pixel values for all models in different symmetric datasets (the number at the top of each histogram is the mean perturbation L2 over all test samples).

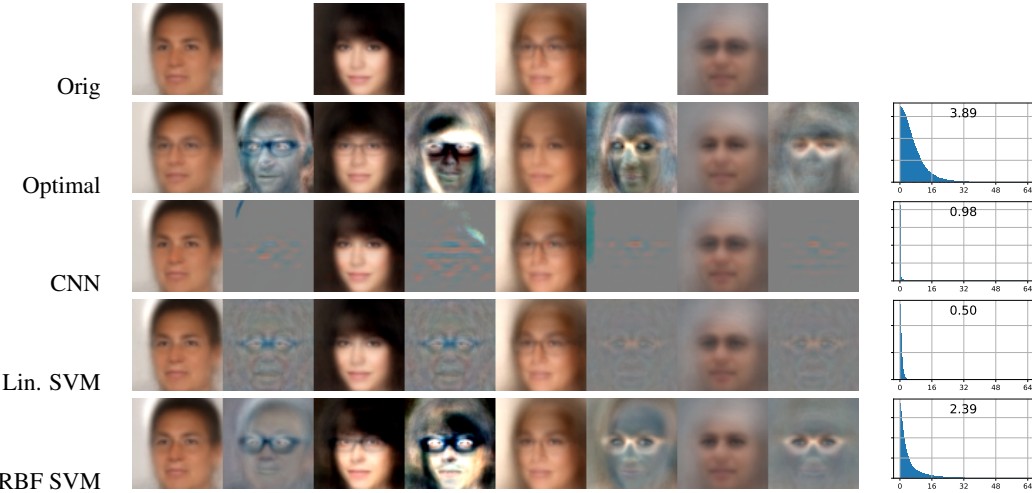

Figure 14: Samples, perturbations and histograms for CelebA attribute 'Eyeglasses'

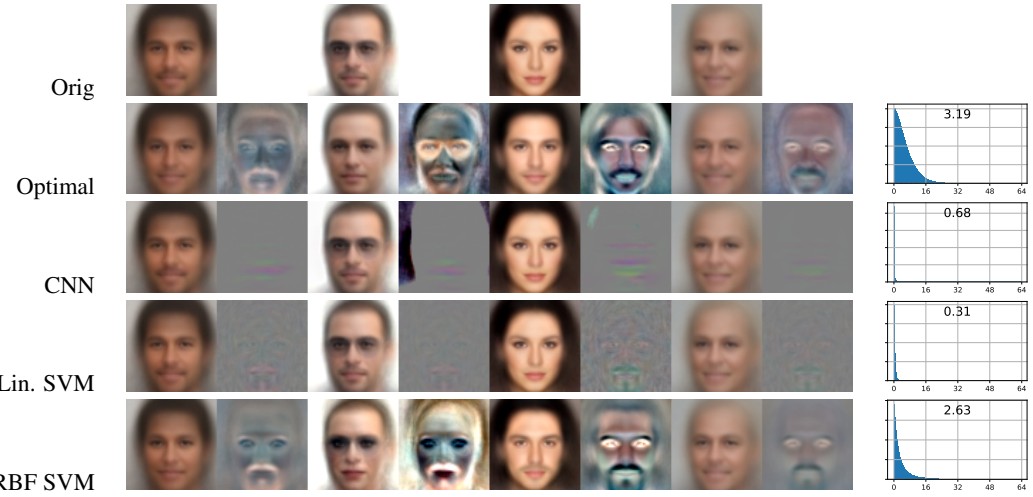

Figure 15: Samples, perturbations and histograms for CelebA attribute 'No Beard'

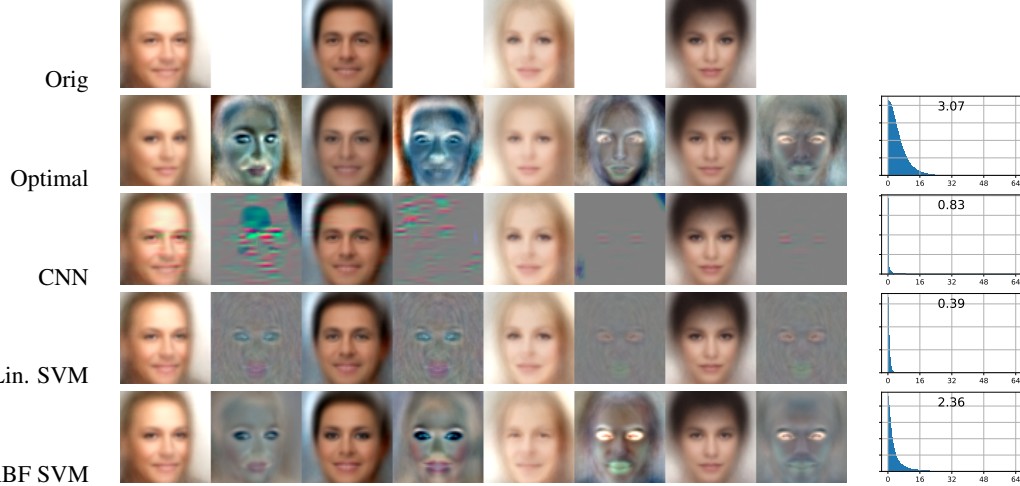

Figure 16: Samples, perturbations and histograms for CelebA attribute 'Heavy Makeup'

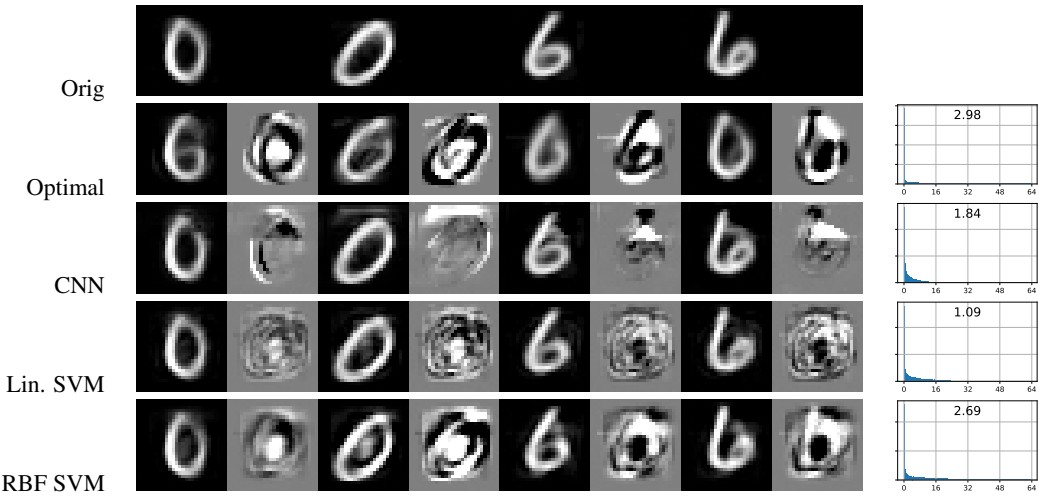

Figure 17: Samples, perturbations and histograms for MNIST digits 0 vs. 6

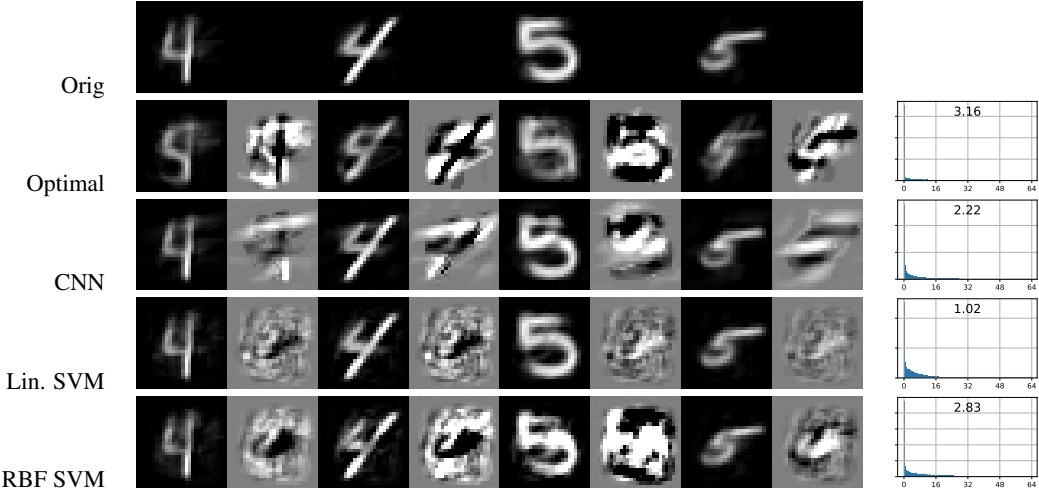

Figure 18: Samples, perturbations and histograms for MNIST digits 4 vs. 5

