# OpenReview forum: "A Bayes-Optimal View on Adversarial Examples"
_ICLR.cc/2020/Conference — Reject_

### Official Review · AnonReviewer2 · 2019-10-23
**Official Blind Review #2**

**Rating:** 6

**Review:**

This paper proposes studying adversarial examples from the perspective of Bayes-optimal classifiers. They construct a pair of synthetic but somewhat realistic datasets—in one case, the Bayes-optimal classifier is *not* robust, demonstrating that the Bayes-optimal classifier may not be robust for real-world datasets. In the other case, the Bayes-optimal classifier is robust, but neural networks fail to learn the robust decision boundary. This demonstrates that even when the Bayes-optimal classifier is robust, we may need to explicitly regularize/incentivize neural networks to learn the correct decision boundary.

The contribution of the two datasets (the symmetric and asymetric CelebA) is, in my opinion, an extremely important contribution in studying adversarial robustness and on their own these datasets warrant further study. Previously, all studies of this sort had to be done with small-scale classifiers and simplistic datasets such as Gaussians. The paper also definitively proves that there are realistic datasets where the Bayes-optimal classifier is non-robust, which goes against quite a bit of conventional wisdom in the field and opens up many new paths for research. However, there are a few (in my opinion) critical concerns that currently bar me from strongly recommending acceptance of the paper. I outline these below.

1. Prior work: the paper seems to ignore a plethora of prior work around studying adversarial robustness and understanding its roots. For example, a few very closely related works are as follows:
   - Adversarial examples are not Bugs, they are Features (https://arxiv.org/abs/1905.02175): Ilyas et al (2019) demonstrate that adversarial perturbations are not in meaningless directions with respect to the data distribution, and in fact a classifier can be recovered from a labeled dataset of adversarial examples. While not in conflict with this work, it does closely relate and discuss many of the same issues discussed in this work, so relating them would be fruitful.

   - A Discussion of Adversarial Examples are not Bugs they are Features (https://distill.pub/2019/advex-bugs-discussion/): Nakkiran (2019) actually constructs a dataset (called adversarial squares) where the Bayes-optimal classifier is robust but neural networks learn a non-robust classifier due to label noise and overfitting. Interestingly, they also construct a dataset where they Bayes-optimal classifier is robust and neural networks *do* learn a robust classifier (adversarial squares sans label noise). While I think the datasets presented in this work are much more interesting and certainly more realistic, this work should be put in context.

    - Excessive Invariance causes Adversarial Vulnerability (https://arxiv.org/abs/1811.00401v3): Jacobsen et al offers an explanation for adversarial examples based on the fact that NNs are not sensitive to many task-relevant changes in inputs, which seems to tie in nicely to the discussion in this paper, as under the presented setup the Bayes-optimal classifier will certainly exploit (and be somewhat sensitive) to such changes.

    - Adversarially robust generalization requires more data (https://arxiv.org/abs/1804.11285): Schmidt et al show a setup where many more samples are required for adversarial robustness than for standard classification error. And it seems to have very relevant connections to your work.

    - In general this list is not comprehensive either: there are many relevant connections to the robustness-accuracy tradeoff (https://arxiv.org/abs/1901.08573, https://arxiv.org/abs/1805.12152), and other works.

2. Discussion/interpretation of the results:
    - Sufficient vs necessary: While the experimental design and results are both of very high quality, I am slightly confused about the interpretation of the results. First, if my understanding of the paper is correct, the experiments show that (a) the Bayes-optimal classifier can be non-robust in real-world settings, and (b) even when the Bayes-optimal classifier is robust, NNs can learn a non-robust decision boundary. In particular, (b) indicates that it may be *necessary* to design regularization methods that steer NNs towards the correct decision boundary—it says nothing about whether these regularization methods will be *sufficient*, which the paper seems to suggest, e.g. in the abstract "our results suggest that adversarial vulnerability is not an unavoidable consequence of machine learning in high dimensions, and may often be a result of suboptimal training methods used in current practice." In fact, if real-world datasets end up being like the asymmetric dataset, then the results of this paper would actually indicate the *opposite* of the above statement. It is unclear on what basis one can say that real-world datasets are more like the symmetric case or the asymmetric case. I believe a more measured conclusion (perhaps that we *need* more regularization methods, but even then we may not be able to get perfect robustness and accuracy) would better fit the strong results presented in the paper.

    - CNN vs Linear SVM: I am confused about why we would expect a CNN to be able to learn the Bayes-optimal decision boundary but not the Linear SVM. The paper justifies the adversarial vulnerability of the Linear SVM by arguing that the Bayes-optimal classifier is not in the Linear SVM hypothesis class, which makes sense. The RBF SVM, for small enough bandwidth can express any function and is convex, so no argument needs to be made about its ability to find the Bayes-optimal classifier. For CNNs, however, it is unclear if the Bayes-optimal classifier lies in the hypothesis class (there are "universal approximation" arguments but these usually require arbitrarily wide networks and are non-constructive)—couldn't it be that the CNNs used here is in the same boat as the Linear SVM (i.e. the Bayes-optimal decision boundary is not expressible by the CNN?)

3. Experimental setup:
    - One somewhat concerning (but perhaps unavoidable) thing about the experimental setup is that all the considered datasets are not perfectly linearly separable, i.e. the Bayes-optimal classifier has non-zero test error in expectation, and moreover the data variance is full-rank in the embedded space. This is in stark contrast to real datasets, where there seem to be many different ways to perfectly separate say, dogs from cats, and the variance of the data seems to be very heavily concentrated in a small subset of directions. I am concerned that these properties are what drive the Bayes-optimal classifier for the symmetric dataset to be robust (concretely, if 0.01 * Identity was not added to the covariance matrix of the symmetric model and the covariance was left to be low-rank, then any classifier which was Bayes-optimal along the positive-variance directions would be Bayes-optimal, and could behave arbitrarily poorly along the zero-variance directions, still being vulnerable). This concern does not make the contribution of the symmetric dataset less valuable, but a discussion of such caveats would help further elucidate the similarities and differences of this setup from real datasets.

    - It is unclear if what is lacking from the NN is explicit regularization, or just more data. In particular, with such low-variance directions, at standard dataset sizes the distributions generated here are most likely statistically indistinguishable from their robust/non-robust counterparts (you can see hints of this in the fact that the CNN gets . While completely alleviating this concern may once again be quite difficult/impossible, it could be significantly alleviated by generating training samples dynamically (at every iteration) instead of generating a dataset in one shot and training on it. It would be very interesting to see whether these results differ at all from the one-shot approach here.

4. A suggestion rather than a concern and not impacting my current score: but it would be very interesting to see what happens for robustly trained classifiers on the symmetric and asymmetric datasets.

Overall, this paper is a very promising step in studying adversarial robustness, but concerns about discussion of prior work, discussion of experimental setup, and conclusions drawn, currently bar me from recommending acceptance. I would be more than happy to significantly improve my score if these concerns can be addressed in the revision and corresponding rebuttal.

**Experience Assessment:**

I have published in this field for several years.

**Review Assessment: Checking Correctness Of Derivations And Theory:**

N/A

**Review Assessment: Checking Correctness Of Experiments:**

I carefully checked the experiments.

**Review Assessment: Thoroughness In Paper Reading:**

I read the paper thoroughly.

---

> ### Author Response · Authors · 2019-11-11
> **Response to Reviewer #2**
>
> Thank you for your review. It is a pleasure to read such a detailed and constructive review.
>
> We have updated the paper to reflect your suggested improvements. Specifically:
>
> Prior Work. We agree that all these related works are relevant and not in conflict with our results, and  have updated the text to include them.
>
> Sufficient vs. Necessary. We agree with the comment and have updated the text to more accurately reflect our view that from the Bayes-Optimal perspective there can be two different causes for adversarial vulnerability: asymmetries in the dataset or suboptimal learning.
>
> CNN vs Linear SVM. To check whether the failure of CNNs to learn a robust classifier is due to a lack of expressive power, we systematically increased the number of channels in the model and measured mean L2 as a function of the expressiveness. The results (shown in figure 8) do not show any increase in robustness as expressive power grows. We believe that these experiments, together with the well-known results on the “universal approximation” property of CNNs, strongly suggest that the problem with CNNs is not that they cannot approximate a robust classifier for this task. In particular, note that the RBF robust classifier is a smooth function of the input and the rate of approximation in the universal approximation literature depends on the smoothness. CNNs have been shown to be capable of expressing random functions of more than one million images (Zhang et al. 2017), so expressing a smooth function, such as the RBF classifier or the Bayes-Optimal classifier, seems perfectly reasonable.
>
> Similarities between synthetic datasets and real datasets. We do believe that the synthetic datasets capture much of the variability that appears in the real datasets. An additional experiment we performed (now in the appendix) is to train a CNN on the synthetic data and measure its performance on the real data. The results show that training on synthetic data, generalizes reasonably well to the real data (with approximately 5% drop in accuracy). While we agree that there are many ways to perfectly discriminate a finite dataset of dogs and cats, this is also true for a finite sample from the synthetic dataset.
>
> Increasing the amount of training examples. Following your suggestion, we conducted additional experiments where the data samples were generated dynamically from the MFA model, and measured robustness as a function of the number of examples (figure 8). We did not see any significant improvement in robustness, even when we went as far as 1 Million training examples for a binary classification task.  Note also that the RBF SVM did learn a robust classifier with as little as 10,000 examples for the same binary tasks.
>
> Robust training. We included an additional experiment where we used robust training for the CNN (We used the TRADES method, from Zhang et al. 2019, which was the winner in a recent adversarial training challenge). As shown in the appendix, this gave only a modest improvement in robustness. Still far from the robustness of the Bayes-Optimal classifier (or the RBF SVM). There are of course many different variants of robust training, but this result shows that the problem of effectively optimizing for both accuracy and robustness is much more difficult in CNNs compared to shallow architectures.

---

> > ### Comment · AnonReviewer2 · 2019-11-11
> > **Re: Response to Reviewer #2**
> >
> > I want to first thank the authors for addressing all of the comments I made in the initial review---the revision of the paper indeed looks significantly improved.
> >
> > I have not had a chance to fully look through the revisions, but in the interest of giving the authors a chance to respond to this, I wanted to write some preliminary comments on the revision.
> >
> > - Prior work: The updated manuscript is indeed much better in regards to citing prior work. I have a few additional comments—based on the current writing of the authors, it would seem as though the two most related works to this one are Ilyas et al (2019) and Nakkiran (2019)  (please correct me if I am misinterpreting). That being the case, it seems worth clarifying a few points and elucidating a few connections:
> >     - Nakkiran (2019) should most likely be cited when introducing the construction of “a dataset where the optimal classifier is robust, but CNNs learn a non-robust classifier” (see the “Adversarial Squares” section of that work)
> >     - It is also definitely worth noting that Nakkiran (2019) finds that CNNs *do* learn a robust classifier naturally unless there is both input noise and label noise in the dataset (in order to induce overfitting). It seems that this is an interesting converse result to the findings of this paper.
> >     - A relevant experiment from Ilyas et al (2019) appears to be the fact that one can actually learn information about the decision boundary from adversarial examples, which, though not corroborating nor disproving the thesis of this work, seems relevant enough to mention (in particular in the context of the "Cindy Crawford" example given in the paper, as the results of Ilyas et al would indicate that it cannot be pure overfitting in the standard sense).
> >     - I am not sure about this statement: “The way we formalize this intuition, below, is quite different from that of (Ilyas et al., 2019) and also leads to quite different conclusions.” If I am understanding correctly, then the model proposed in observation 1 is very similar to the one proposed in Ilyas et al (Section 4 of that work considers precisely this two-gaussian setting)
> >
> > New experiments: I appreciate that the authors have put in effort to run experiments regarding my earlier comments, they have greatly clarified things. I still have a few concerns:
> >     - Is there a graph somewhere of (# data points) vs (robust accuracy)? I can't seem to find it in the revision.
> >     - In order to make the symmetric dataset, it seems that samples are generated according to a gaussian with covariance $AA^T + \sigma * I$. Is there a good understanding of how the $\sigma$ is affecting robustness here?
> >     - I am slightly confused at the aforementioned discrepancy between this work and that of Nakkiran (2019). In particular, there the author finds that training a neural network on a simple gaussian dataset does in fact yield a significantly robust classifier. I think understanding the role of \sigma may play a big role here, but it would be nice for the authors to provide some context.
> >
> > Finally, I think the authors may have misinterpreted my comment about the similarity between this dataset and real datasets. In particular, my point is not that there are infinite ways to separate a finite sample of dogs and cats, but rather that even in the infinite-data setting, there is not one unique classifier that can perfectly distinguish pictures of dogs from pictures of cats. In contrast, by adding the $0.001 * I$, you ensure that there is only one Bayes-optimal classifier (and that it is robust).
> >
> > More explicitly, my concern in this respect is that the $0.001 * I$ being added to the covariance is “cheating,” by making it so that there is a unique Bayes-Optimal Classifier, when in reality, if this 0.001 was 0, there would be infinite Bayes-Optimal classifiers, many of which would not be robust. My concern is that this 0.001 is too small for any reasonable learning algorithm to pick up on—in other words, in practice, this 0.001 is a 0, in which case there is no unique Bayes-Optimal classifier, and many of them are non-robust—it thus makes sense that unless your learning algorithm is explicitly margin-maximizing, you end up choosing one of these optimal classifiers that is non-robust.
> >
> > I am happy to expand on this further if my explanation above does not make sense. Also note that this concern does not devalue the experiments or invalidate any results in and of itself, but I think it deserves highlighting and discussing in the context of the paper.

---

> > > ### Author Response · Authors · 2019-11-12
> > > **Re: Re: Response to Reviewer #2**
> > >
> > > Thank you again for these detailed and constructive comments.
> > >
> > > Prior works. We agree that the most closely related work should be discussed more and have modified the version accordingly. Note that the analysis in section 4 of  Ilyas et al. is for linear classifiers (or equivalently Gaussians with the same covariance for the two classes). When the covariance of the Gaussians is different between the two classes, the Bayes Optimal classifier is nonlinear, so observation 1 is already quite different from the analysis in Ilyas et al. (and of course observations 2,3,4 are even more different). We have also discussed Nakkiran’s construction in section 4 (along with a very similar construction that appeared in Tanay and Griffin).
> > >
> > > Experiments:
> > >
> > > The graph of data points vs robustness is in the appendix of the revised version (figure 8, left).
> > >
> > > Indeed in a mixture of factor analyzers (MFA) model, each component has covariance which is a sum of a low rank matrix (this models the covariance on the local manifold of images) plus a diagonal matrix (this measures the covariance outside the manifold). Without the diagonal component, the covariance is not full rank and hence is not a valid covariance matrix. In order for an MFA model to produce realistic images, the sigma needs to be small (once sigma is above 0.03 approximately, the generated images appear quite noisy and unrealistic). We have now added additional experiments (figure 9) with ranges of sigma ranging from 0.001 (as was used in the original version) and up to 0.05. There does seem to be a small increase in robustness of the learned CNN but it is still very brittle and far from the robustness of the Bayes-Optimal. Note also that increasing sigma is equivalent to simply adding Gaussian IID noise to each pixel of the training examples, which has previously been reported to give a small increase in robustness of CNNs, but is usually considered to be inferior to adversarial training.
> > >
> > > Regarding the “dogs” vs. “cats”, we think this is perhaps a philosophical argument. In our view, for most realistic classification experiments, there is a unique Bayes Optimal classifier that would maximize accuracy. We agree that when there are regions of the input space that have exactly zero probability under both classes, then the Bayes-Optimal classifier is not unique, but for images that are measured with physical cameras, that seems to us highly implausible (and if it were true, then adversarial attacks should be restricted to not use these zero probability parts of the space). We do agree with the concern that the sigma we used was so small as to be effectively zero, but as noted above, our results still hold with a sigma that is 50 times larger, where the noise is visible even in a single image.
> > >
> > > Regarding Nakkiran’s squares, we believe it is consistent with our discussion at the end of the paper: several authors have pointed out that CNNs trained using SGD appear to have some implicit form of regularization that favors large margins, but the strength of this regularization depends in a very complicated manner on the exact details of the training. For simple toy problems, even a tiny amount of regularization may be enough to ensure robustness, and we believe this is the case with Nakkiran’s squares (where a linear SVM would already be robust). But for more realistic and complicated  datasets, such as the ones that we have presented in our paper,  the amount of  regularization that is given by “standard” CNN training is not enough, and the challenge is to develop explicit regularization methods that would allow us to efficiently explore accuracy and robustness in such datasets.

---

> > > > ### Comment · AnonReviewer2 · 2019-11-12
> > > > **Thanks**
> > > >
> > > > Prior work: Thank you, I will take a closer look at the revision but at first glance it looks like a much better contextualization of the work.
> > > >
> > > > I also appreciate the clarification and addition of various experiments, and the connection between adversarial training and noise training is a good point.
> > > >
> > > > My only disagreement is now with regards to the "philosophical discussion" cited by the authors. It think it's rather clear that many non-trivial subsets of R^d are assigned exactly 0 probability by any reasonable generative model over, say, dogs and cats. Again, finding perfect generative models of the world is of course beyond the scope of this work which already makes a valuable contribution, but I think it could be valuable to briefly mention this topic in a revision.
> > > >
> > > > One experiment that might actually quantify the extent of this issue is to project all the images to the subspace with non-trivial variance (the space defined by the matrix A). I would be curious to see what the robustness of a CNN trained on this dataset would look like.
> > > >
> > > > I am, however, already raising my score since the authors have done well to address and run experiments around all my concerns.

---

> > > > > ### Author Response · Authors · 2019-11-15
> > > > > **Additional response to Reviewer #2**
> > > > >
> > > > > Thank you again for your comments and suggestions.
> > > > >
> > > > > Regarding the suggested experiments with projecting all images down to a linear subspace, note that in the MFA model there are many linear subspaces, and different regions of $R^d$ would need to be projected to a different subspace, so this is a complicated idea to try.  We have added a figure (figure 8) in the appendix to better illustrate the geometry of MFA models, and the roles of $A$ and $\sigma$.
> > > > > We have also added a new experiment with real data, suggesting that the robustness of different classifiers on real data is similar to what we observed in the synthetic data. In particular, for the real Male/Female data, CNN and Linear SVMs learn a very vulnerable classifier, while RBF SVM learns a robust classifier whose adversarial examples are perceptually meaningful.

---

> > > > > > ### Comment · AnonReviewer2 · 2019-11-15
> > > > > > **Clarification**
> > > > > >
> > > > > > Is there an analog of Figure 6 for the asymmetric dataset? It would be interesting to see what the RBF SVM does on this dataset. (I understand it's the last day of rebuttals though so this might be a bit late of a question to ask).

---

### Official Review · AnonReviewer3 · 2019-10-24
**Official Blind Review #3**

**Rating:** 1

**Review:**

The paper analyzed the adversarial examples from the Bayes-optimal view. Specifically, the authors analyzed the relationship between the symmetry of covariance of data distribution and the amount of data which are close to the decision boundary. The authors proved that when the covariance of data distribution is asymmetric, a large amount of data will be close to the decision boundary (easy to be attacked). The authors also provided the new datasets which is easy to compute for the bayes-optimal classifier so as to verify the effect of symmetry of covariance on vulnerability of classifier. Moreover, the paper indicated that the vulnerability of CNNs is due to asymmetric distributions or non-optimal learning.

It is interesting that the paper investigated the adversarial examples from the Bayes-optimal view. However, there are some drawbacks:

1.	The motivation of this paper is not clear to me. In other words, what is the benefit of analyzing the adversarial examples from the Bayes-optimal viewpoint, since Bayes model mentioned in this paper is easy to attack. I am not fully convinced by the presentation of the paper.

2.	The theorem or the observation in the paper appears too straightforward. And the ‘observation 1’is not general. The authors may need to consider more general cases that when the standard deviation of eigen value of covariance matrix is large, the Bayes model will be easily attacked. (not just the case that one of eigen value is zero).

3.	One minor point, it appears somewhat strange that “observations” were proved. It is better to change observations to theorems or lemmas.

4.	The authors tried to explain directly the vulnerability of CNN in a same way. However, CNN is a totally different model compared with the Bayes model (one is a discriminative model and the other is a generative model). For generative models, the classification boundary is closely related to all training samples. Therefore, the variance of data distribution is important for attack. For discriminative models, the decision boundary is related to local information. It may not be proper to analyze CNN in the way same as the Bayes model. This should be further clarified and discussed.


**Experience Assessment:**

I have published in this field for several years.

**Review Assessment: Checking Correctness Of Derivations And Theory:**

I carefully checked the derivations and theory.

**Review Assessment: Checking Correctness Of Experiments:**

I carefully checked the experiments.

**Review Assessment: Thoroughness In Paper Reading:**

I read the paper thoroughly.

---

> ### Author Response · Authors · 2019-11-11
> **Response to Reviewer #3**
>
> Thank you for reading our paper. We hope we can clarify (below) some misunderstandings and hope that you will read it again in the light of our comments.
>
> “What is the benefit of analyzing the adversarial examples from the Bayes-Optimal viewpoint, since Bayes model mentioned in this paper is easy to attack.” Actually,  as we show in the paper, the Bayes-Optimal classifier may be easy or hard to attack depending on the data distribution. More generally, adversarial examples present an intriguing and complex phenomena, and we believe that analyzing them from the Bayes-Optimal perspective allows us to disentangle the different possible causes.
>
> We didn’t fully understand the comment that “observation 1 is not general”. Specifically, we do not require zero eigenvalue for the vulnerability condition to occur (but asymmetry in the variance).
>
> “CNN is a totally different model compared with the Bayes model (one is a discriminative model and the other is a generative model)”. Actually the Bayes-Optimal model (as its name suggests) gives the best possible accuracy in the discrimination task. Thus if the CNN wants to optimize the accuracy, then it should agree with the Bayes-Optimal model.

---

> > ### Comment · AnonReviewer3 · 2019-11-14
> > **Further remarks...**
> >
> > The reviewer appreciates the response from the authors.
> >
> > In general,  the paper illustrated interesting observations that asymmetric distributions could lead to the high vulnerability. This could be easy to understand. Frankly speaking,  these observations seem straightforward as the optimal decision plane would be biased towards the direction with the minimum variance. So, though this  is interesting and explicitly pointed out by the paper, I would not deem this a significant contribution to the adversarial learning.
> >
> > The question is, it still remains uncertain why Deep learning could be brittle.  In the first part, the paper showed that a Bayes classifier could be brittle due to the asymmetric distribution, which is again easy to understand; then in the second half,  it is implied that the vulnerability of CNN may be due to the sub-optimal model itself.  I would agree on both the points, which are however straightforwardly and not a surprise to the reviewer. Anyway,  Aren't data or model  the only two factors that would affect the training of any algorithms? On the other hand, it would be more interesting if the authors can analyze what would the distribution look like for the data features in the last hidden layer, since the distribution in the feature space would be more relevant to the decision of DNN.
> >
> > Moreover, in practice, the data are not as simple as a mixture of Gaussian， it might be also more complicated to reach a firm conclusion only by inputting into CNN data following such simple distributions.  Anyway, CNN or other discriminative classifiers （e.g. SVM) did not try to find a Bayes-optimal decision boundary (even if the distributions can be known). Another point is, the problem encountered by the asymmetric distribution may be allevaited  by certain regularization or normalization, which the CNN can usually adopt.  How would be the robustness after this has been done? All these points may be more interesting to be further elaborated and/or discussed.
> >
> > In summary, it is a piece of interesting work and also be somewhat inspiring, but  I have to say, to the reviewer, this paper is in lack of necessary depth and contain no highly-valuable insight.

---

> > > ### Author Response · Authors · 2019-11-15
> > > **Additional response to Reviewer #3**
> > >
> > > We appreciate the clarifications about the reviewer’s concerns.
> > >
> > > The sentence “Frankly speaking,  these observations seem straightforward as the optimal decision plane would be biased towards the direction with the minimum variance. “  seems to suggest a misunderstanding. Our analysis is for general classifiers (not linear classifiers) and the directions of the adversarial examples are not necessarily in a direction of minimum variance. The situation with general classifiers is more subtle than the sentence suggests, and by understanding this situation we have been able to construct realistic image datasets where all the points are arbitrarily close to the decision boundary for an optimal, nonlinear classifier.  Note also that our analysis is not just for a mixture of Gaussians (see observations 3 and 4).
> > >
> > > Regarding “Aren't data or model  the only two factors that would affect the training of any algorithms? “.  Note that many authors have suggested that high dimensionality is critical for adversarial examples, and we have shown that this is not a necessary nor a sufficient factor for the presence of adversarial vulnerability. We have also suggested a third factor (expressive power) that explains why the linear SVM fails to find a robust classifier while the RBF SVM does.
> > >
> > > We are not sure we understand how analyzing the distribution of some deep layer features can help, since due to the strong non-linearity, the features could be separable with a large margin while at the same time the input images (in the original image space) might not be.
> > >
> > > Regarding the relevance of our symmetric dataset to real data, we added an experiment (Table 2) that shows that CNNs trained on the symmetric data can classify real test images (with some expected degradation in accuracy). We also trained the three models (CNN, Linear and RBF SVM) on real CelebA Male/Female data and got similar results to our symmetric data experiment (i.e. CNN and linear SVM are vulnerable and RBF SVM is robust), see figure 12.
> > >
> > > Regarding robustified CNNs, we added an experiment in which we train the CNN using state-of-the-art adversarial training method and got only a small improvement in robustness (figure 10).
> > >
> > > We hope that you will read the revised version and our clarifications and see if it changes your opinion.

---

### Official Review · AnonReviewer1 · 2019-10-27
**Official Blind Review #1**

**Rating:** 3

**Review:**

The paper studies the adversarial robustness of the Bayes-optimal classifier (i.e., optimal for the standard "benign" risk). To do so, the authors construct various synthetic distributions and show that in some cases the Bayes-optimal classifier is also adversarially robust, while in other cases it is not. In the main experiment, the authors construct two high-dimensional synthetic distributions of human faces via a generative model. In one of the distributions, even the Bayes-optimal classifier is vulnerable to adversarial examples. In the other distribution (where the Bayes-optimal classifier is robust), CNNs do not achieve high robustness while other approaches such as an RBF SVM are more robust.

Overall I find the experiment in the paper interesting, but it is unclear how representative the experiments are for adversarial robustness on real data. There are natural follow-up experiments that would shed some light on this question and could substantially strengthen the paper. Hence I unfortunately recommend to reject the paper at this point and encourage the authors to deepen their experimental investigation. For instance, the following points would be relevant:

- Does adversarial training / robust optimization result in a robust neural network on the synthetic data distribution where the Bayes-optimal classifier is robust (and the RBF SVM is more robust than a CNN)?

- Do RBF SVMs also exhibit higher adversarial robustness than CNNs on comparable real datasets? This would indicate to what extent the synthetic distributions are representative of real data w.r.t. adversarial robustness.


Additional comments:

- Briefly defining the MFA model in the main text would provide helpful context.

- A few more details about the experiments could be informative in the main text, e.g., the CNN architecture and the accuracies the various methods achieve.

- An end-of-proof symbol at the end of proofs would be helpful to the reader.

- Is there an index i missing in \pi in Equation (8)?

- Is the probability given in (9) exact? Gaussians are supported on all of R^d, so even a Gaussian component far away will contribute to the probability of a point under p_1, at least a (very) small amount.

- The proof of Observation 2 is more a sketch. It would be good to include a more formal proof in the appendix.

**Experience Assessment:**

I have published in this field for several years.

**Review Assessment: Checking Correctness Of Derivations And Theory:**

I assessed the sensibility of the derivations and theory.

**Review Assessment: Checking Correctness Of Experiments:**

I assessed the sensibility of the experiments.

**Review Assessment: Thoroughness In Paper Reading:**

I read the paper at least twice and used my best judgement in assessing the paper.

---

> ### Author Response · Authors · 2019-11-11
> **Response to Reviewer #1**
>
> Thank you for your review. We have uploaded a new version with additional experiments based on your suggestions. We hope you take a second look at the paper and see if it changes your recommendation.
>
> Regarding your major comments:
>
> Adversarial training. We included an additional experiment where we used robust training for the CNN (We used the TRADES method, from Zhang et al. 2019, which was the winner in a recent adversarial training challenge). As shown in the appendix, this gave only a modest increase in robustness. Still far from the robustness of the Bayes-Optimal classifier (or the RBF SVM). There are of course many different variants of robust training, but this result shows that the problem of effectively optimizing for both accuracy and robustness is much more difficult in CNNs compared to shallow architectures.
>
> Similarities between synthetic datasets and real datasets. We do believe that the synthetic datasets capture much of the variability that appears in the real datasets. An additional experiment we performed (now in the appendix) is to train a CNN on the synthetic data and measure its performance on the real data. The results show that training on synthetic data, generalizes reasonably well to the real data (with approximately 5% drop in accuracy).
>
> We focused on experiments with synthetic datasets because they allow us to systematically explore the different possible causes of adversarial examples (for example, having a robust optimal classifier indicates that the cause of the vulnerability of a trained model does not lie in the data). While we agree that the robustness of RBF SVMs on real data is interesting to pursue, this is not the main focus of our work, which aims to disentangle the different causes of adversarial vulnerability.
>
> We have also changed the manuscript in light of your “additional comments” (e.g. we added the missing index i in equation 8).  Regarding equation 9, if $g_{ik}(x)$ is a delta function (as is assumed above equation 9) then equation 9 is exact (see e.g. Neal and Hinton 98).

---

> > ### Author Response · Authors · 2019-11-15
> > **Additional response to Reviewer #1**
> >
> > Thank you for your comments.
> >
> > Please note that in the final version, we have added an additional experiment following your suggestion. Specifically, we train CNNs, Linear SVMs and RBF SVMs on real data. We find a similar pattern in the real data that we found in our synthetic data.  In particular, for the real Male/Female data, CNN and Linear SVMs learn a very vulnerable classifier, while RBF SVM learns a robust classifier whose adversarial examples are perceptually meaningful (figure 12).

---

### Author Response · Authors · 2019-11-15
**Significantly Revised Version**

We thank the reviewers for their comments and interaction during the rebuttal phase. We encourage all of them to read the latest version, which has been significantly revised following their comments. Specifically we have improved the discussion of related work, clarified the conclusions and added details about the MFA model. Perhaps more importantly, we have added new experiments including adversarial training of CNNs and experiments on real data. Overall we believe the new version supports the conclusions in our previous version but using stronger experimental evidence and more context.

---

### Author Response · Authors · 2020-02-05
**Revised version**

We believe we have fully addressed the reviewers' concerns in the revised paper. Unfortunately, as the meta-review  states: "I am not certain that they reviewed the revision, despite my prodding".

A revised and improved version of this paper is now submitted to a different venue.

The one reviewer who did read the revised version wrote: "The contribution of the two datasets is, in my opinion, an extremely important contribution in studying adversarial robustness."
"Overall, this paper is a very promising step in studying adversarial robustness".

---

### Decision · Program_Chairs · 2019-12-19

**Decision:**

Reject

**Comment:**

The paper studies how adversarial robustness and Bayes optimality relate in a simple gaussian mixture setting. The paper received two recommendations for rejection and one weak accept. One of the central complaints was whether the study had any bearing on "real world" adversarial examples. I think this is a fair concern, given how limited the model appears on the surface, although perhaps the model is a good model of any local "piece" of a decision boundary in a real problem. That said, I do not agree with the strong rejection (1) in most places. The weak reject asked for some experiments. The revision produced these experiments, but I'm not sure how convincing these are since only one robust training method was used, and it's not clear that it's the best one could do among SOTA methods. For whatever reason, the reviewers did not update their scores. I am not certain that they reviewed the revision, despite my prodding.